# Invasive Aspergillosis with Cavernous Sinus Thrombosis Following High-Dose Corticosteroid Therapy: A Challenging Case of Rhino-Orbital-Cerebral Mycosis

**DOI:** 10.3390/jof10110788

**Published:** 2024-11-13

**Authors:** Faruk Karakeçili, Orçun Barkay, Betül Sümer, Umut Devrim Binay, Kemal Buğra Memiş, Özlem Yapıcıer, Mecdi Gürhan Balcı

**Affiliations:** 1Faculty of Medicine, Infectious Diseases and Clinical Microbiology Department, Erzincan Binali Yıldırım University, Erzincan 24100, Turkey; drfarukkarakecili@hotmail.com (F.K.); o.barkay1985@gmail.com (O.B.); drbetulsumer@gmail.com (B.S.); devrimbinay@hotmail.com (U.D.B.); 2Faculty of Medicine, Radiology Department, Erzincan Binali Yıldırım University, Erzincan 24100, Turkey; kemalbugramemis@gmail.com; 3Medical Pathology Department, Bahçeşehir University Medicine Faculty Hospital, İstanbul 34349, Turkey; ozlemyapicier@yahoo.com; 4Faculty of Medicine, Pathology Department, Erzincan Binali Yıldırım University, Erzincan 24100, Turkey

**Keywords:** aspergillosis, cavernous sinus thrombosis, corticosteroids, COVID-19, fungal sinusitis

## Abstract

Invasive aspergillosis is a rare but severe fungal infection primarily affecting immunocompromised individuals. The *Coronavirus* Disease-2019 (COVID-19) pandemic has introduced new complexities in managing aspergillosis due to the widespread use of corticosteroids for treating COVID-19-related respiratory distress, which can increase susceptibility to fungal infections. Here, we present a challenging case of progressive cerebral aspergillosis complicated by cavernous sinus thrombosis (CST) in a 67-year-old male with a history of COVID-19. The patient, initially misdiagnosed with temporal arteritis, received pulse corticosteroid therapy twice before presenting with persistent left-sided headaches and vision loss. Cranial imaging revealed findings consistent with fungal sinusitis, Tolosa–Hunt syndrome, and orbital pseudotumor, which progressed despite initial antifungal therapy. Subsequent magnetic resonance imaging indicated an invasive mass extending into the left cavernous sinus and other intracranial structures, raising suspicion of aspergillosis. A transsphenoidal biopsy confirmed *Aspergillus* infection, leading to voriconazole therapy. Despite aggressive treatment, follow-up imaging revealed significant progression, with extension to the right frontal region and left cavernous sinus. The patient then developed visual impairment in the right eye and was diagnosed with CST secondary to fungal sinusitis. Management included a combination of systemic antifungals and antibiotics; however, the patient declined surgical intervention. This case underscores the diagnostic challenges and rapid progression associated with cerebral aspergillosis in post-COVID-19 patients treated with corticosteroids. This report highlights the need for heightened clinical suspicion and prompt, targeted interventions in similar cases to improve patient outcomes. Further research is required to understand the optimal management of invasive fungal infections.

## 1. Introduction

Aspergillosis is a fungal infection caused by *Aspergillus* species, a mold commonly found in soil, decomposing organic matter, and indoor environments such as air conditioning systems. *Aspergillus* species produce airborne conidia (spores) small enough to be inhaled into the respiratory tract. While most individuals encounter *Aspergillus* spores daily without any clinical consequences, immunocompromised patients or those with underlying pulmonary conditions are at increased risk of developing invasive aspergillosis. Invasive forms primarily affect the lungs, but dissemination to the brain, sinuses, and other organs can occur, posing significant clinical challenges. Early diagnosis and aggressive treatment are crucial for improving outcomes, especially in complex cases involving corticosteroid use, as seen in patients recovering from *Coronavirus* Disease-2019 (COVID-19). The COVID-19 pandemic has significantly altered the landscape of opportunistic fungal infections. Corticosteroid use, a cornerstone in managing COVID-19-induced acute respiratory distress syndrome, has been linked to an increased susceptibility to fungal infections, particularly invasive pulmonary aspergillosis. COVID-19-associated pulmonary aspergillosis has increasingly been recognized as a complication in critically ill COVID-19 patients. Aspergillosis can present in various forms, with a wide range of clinical manifestations [1,2,3,4,5]. Invasive aspergillosis is a severe and potentially life-threatening infection primarily affecting immunocompromised individuals, especially those who have undergone organ or stem cell transplantation. The most frequently implicated *Aspergillus* species are *Aspergillus fumigatus* and *A. flavus*, with less common species including *A. terreus*, *A. nidulans*, *A. niger*, and *A. versicolor* [6,7]. Although *Aspergillus* primarily affects the lungs, in rare and severe cases it can disseminate to other organs, including the central nervous system, leading to cerebral aspergillosis. Invasive cerebral aspergillosis is a life-threatening condition often associated with high mortality. It is characterized by the infiltration of fungal hyphae into the brain tissue, leading to complications such as abscess formation, meningitis, or vascular involvement including cavernous sinus thrombosis (CST) [8]. The transmission of *Aspergillus* typically occurs through the inhalation of airborne conidia, with nosocomial infections arising sporadically in hospital settings, particularly during construction or renovation activities that disturb spore-containing dust. A definitive diagnosis generally requires isolating the organism from a sterile site and histopathological confirmation of tissue invasion. Additional diagnostic tools include radiological imaging, galactomannan antigen detection, Beta-D-glucan assays, and polymerase chain reaction (PCR) tests [9,10,11]. First-line therapy for invasive aspergillosis is voriconazole, with alternative treatments including lipid formulations of amphotericin, posaconazole, isavuconazole, itraconazole, caspofungin, and micafungin [12]. Here, we present a case of invasive cerebral aspergillosis secondary to corticosteroid therapy administered for COVID-19, and consequently for temporal arteritis, which was a clinical misdiagnosis.

The case describes a 67-year-old male with a history of recent COVID-19 who presented with intermittent left-sided headaches and vision loss in the left eye. The patient received one gram corticosteroid treatment for COVID-19 and pulse steroid treatment with a preliminary diagnosis of temporal arteritis but was later diagnosed with fungal sinusitis, Tolosa–Hunt syndrome, and orbital pseudotumor. The patient was treated with amphotericin B and underwent functional endoscopic sinus surgery; however, follow-up imaging revealed worsening findings. The patient underwent a transsphenoidal biopsy, and the pathology report indicated fungal infection consistent with aspergillosis. The patient was then treated with voriconazole and underwent multiple imaging studies, which showed a significant progression of the lesion in the right frontal region. Surgery was recommended, but the patient declined and continues to be followed up as an outpatient.

The complexity of this case lies in the diagnostic delay and the rapid progression of the infection to the central nervous system, leading to CST and optic nerve involvement.

## 2. Case Report

The case was managed and documented at Erzincan Binali Yıldırım University Mengücek Gazi Training and Research Hospital, with the consent of the patient obtained prior to reporting. Data for this case report were collected retrospectively from patient records, including diagnostic imaging, laboratory tests, and therapeutic interventions. No statistical analysis was required due to the nature of the study.

A comprehensive diagnostic workup was performed, including the following:Magnetic Resonance Imaging (MRI): Cranial and orbital MRI were used as primary imaging modalities to assess the presence of inflammation, masses, and other abnormalities.Computed Tomography (CT) Scans: Complementary CT scans were performed to visualize the extent of sinus involvement and assess for potential bone erosion or invasion of surrounding structures.Biopsy and Histopathological Examination: A transsphenoidal biopsy of the mass lesion was performed, with histopathological staining (e.g., PAS staining) used to confirm the presence of fungal elements consistent with aspergillosis.Microbiological cultures were obtained from biopsy specimens. Fungal cultures such as Sabouraud dextrose agar were used. In addition, laboratory investigations also included acute phase reactants.

The patient was managed by a multidisciplinary team involving specialists in infectious diseases, neurology, ophthalmology, radiology, pathology, and neurosurgery. Each department contributed to the diagnosis and treatment planning, and the patient underwent several consultations:Infectious disease specialists oversaw the antifungal therapy, determining the appropriate drug regimen.Radiologists monitored the progression of lesions through serial MRI and CT scans.Neurosurgeons were involved in evaluating the need for surgical intervention due to the progression of CNS involvement.Pathologists evaluated the pathology preparations.

A 67-year-old male patient, with a history of hypertension and coronary artery disease, contracted COVID-19 in March 2021. After recovering, he began experiencing intermittent, throbbing headaches localized to the left eye and frontotemporal region. By May 2021, he developed vision loss in the left eye, leading to a preliminary diagnosis of temporal arteritis at our neurology clinic, for which he received pulse corticosteroid therapy.

Despite a 10-day course of corticosteroids, the patient’s symptoms persisted, and he sought care at an external center in June 2021, where he again received pulse corticosteroids. A cranial MRI at that time showed findings consistent with fungal sinusitis, Tolosa–Hunt syndrome, and orbital pseudotumor (Figure 1).

He was initially treated with amphotericin B, as the ear, nose, and throat (ENT) team did not suspect invasive fungal sinusitis and did not recommend surgical intervention. After completing a 21-day course of amphotericin B, follow-up cranial MRI in July 2021 revealed progression of the sinus findings. He then underwent functional endoscopic sinus surgery, was discharged, and amphotericin B was discontinued.

In December 2021, the patient’s symptoms recurred. An MRI revealed a heterogeneously enhancing extra-axial mass lesion measuring 41 × 37 mm, which infiltrated the intracranial segment of the left optic nerve, extended to the left parasellar cistern, surrounded the supraclinoid segment of the internal carotid artery, and filled the left cavernous sinus. The differential diagnosis included trigeminal nerve schwannoma (Figure 2A,B).

A positron emission tomography–computed tomography (PET-CT) scan detected no other pathological findings, suggesting the lesion was a primary pathology. A transsphenoidal biopsy confirmed fungal infection compatible with aspergillosis, as indicated by two pathologists, one from an external center (Figure 3), although no growth was observed in the fungal culture.

He was started on voriconazole for two weeks. Following this, an ENT consultation recommended surgery, which the patient declined, and he was discharged on oral voriconazole therapy. In January 2022, a follow-up contrast-enhanced cranial and orbital MRI showed significant progression compared to previous findings. More pronounced sinus changes were observed in the left sphenoid sinus, with contrast enhancement affecting the entire left cavernous sinus, extending anteriorly to the left frontobasal bone and dura mater, encircling the left temporal lobe and extending to the right cavernous sinus (Figure 2C,D). In February 2022, the patient developed visual impairment in the right eye and was diagnosed with CST secondary to fungal sinusitis. He had been on oral voriconazole treatment for approximately 1.5 months and was subsequently treated with intravenous (IV) cefepime, vancomycin, and voriconazole. Surgery was not recommended at that time. After approximately 14 days of treatment, he was discharged, with MRI showing reduced contrast enhancement in the lesion compared to previous imaging. He was referred to an ophthalmologist, and a visual evoked potential test was performed, revealing a prolonged latency of 129 ms and delayed responses in the right eye. After consultation with ophthalmology and neurology, no additional interventions were recommended beyond continuation of aspergillosis treatment.

During outpatient follow-up, in August 2022, a control contrast-enhanced cranial MRI revealed a 4 cm T2A hyperintense, T1A hypointense lesion with peripheral contrast enhancement, intense edema, and sulcal effacement in the right frontal region (Figure 4).

He was again advised to undergo surgery by neurosurgery, but he declined. He continues to receive long-term oral voriconazole therapy and remains under close outpatient follow-up.

## 3. Discussion

Fungal sinusitis is a rare but potentially life-threatening infection caused by various fungi, including *Aspergillus* species, which primarily affects immunocompromised individuals and those with underlying conditions such as diabetes mellitus, malignancy, and chronic steroid use [13,14,15]. The patient in this case had a history of hypertension and coronary artery disease—known risk factors for severe COVID-19 infection [16,17,18]—and was initially misdiagnosed with temporal arteritis, leading to two courses of pulse steroid treatment.

The role of corticosteroids in exacerbating fungal infections is well documented [19]. Particularly in COVID-19 patients, corticosteroid therapy suppresses the immune response, increasing susceptibility to opportunistic infections like invasive aspergillosis [20]. In this case, repeated courses of corticosteroid treatment likely facilitated the spread of *Aspergillus* into the cranial cavity, as noted in other post-COVID-19 patients treated with corticosteroids.

The diagnosis of fungal sinusitis can be challenging as it presents with non-specific symptoms such as headache, facial pain, and nasal congestion, which can mimic other conditions like migraine, sinusitis, and dental problems [21,22]. In this case, the patient initially received pulse steroid treatment with a preliminary diagnosis of temporal arteritis, which is an inflammatory condition affecting the blood vessels that supply the head and neck region [23,24]. However, the persistence of symptoms and the imaging findings prompted further evaluation, leading to the diagnosis of fungal sinusitis. Imaging played a critical role in tracking disease progression, with MRI scans revealing not only the infection’s persistence but also its spread into intracranial structures. While imaging is valuable, a definitive diagnosis of cerebral aspergillosis requires histopathological confirmation, as shown by the transsphenoidal biopsy in this case. Pathological findings, including the identification of septate hyphae typical of *Aspergillus*, confirmed the diagnosis. Other studies have reinforced the importance of combining imaging findings with invasive diagnostic procedures such as biopsy, especially when non-invasive methods like galactomannan antigen detection or PCR fail to provide a definitive diagnosis [25]. Early use of these diagnostic modalities could potentially lead to earlier treatment and better outcomes.

Treatment for fungal sinusitis involves a combination of antifungal therapy and, in some cases, surgical intervention, depending on the infection severity and the disease extent [26,27,28]. In this case, the patient received amphotericin B therapy and underwent functional endoscopic sinus surgery. However, the follow-up imaging studies revealed the progression of findings, and the patient underwent a transsphenoidal biopsy, which confirmed the diagnosis of fungal infection compatible with aspergillosis. The patient was subsequently treated with voriconazole, one of the first-line antifungal agents that is effective against *Aspergillus* species [29,30]. The patient’s refusal of surgery presents a challenge commonly encountered in clinical practice, where patient autonomy must be balanced with medical recommendations.

The progression of the lesion seen in the right frontal region on imaging studies suggests a potential dissemination of the fungal infection. CST is a rare but potentially life-threatening complication of fungal sinusitis. Infection can spread from the paranasal sinuses to the cavernous sinus, leading to inflammation and thrombosis that can lead to visual impairment, seizures, and even death [31,32,33]. Managing CST in the context of fungal infections is particularly challenging, often requiring a combination of aggressive antifungal therapy and, in some cases, anticoagulation because of the risk of thrombosis. However, there is no consensus on anticoagulant use in fungal CST due to the increased risk of bleeding associated with the combination of fungal invasion and anticoagulation [34]. In this case, the patient was treated with voriconazole and other systemic antifungals without surgical intervention, which highlights the limitations of current management options for patients who refuse or are not candidates for surgery.

The prognosis for cerebral aspergillosis is generally poor, particularly when complicated by CST [35]. Despite treatment with voriconazole and IV antibiotics, the patient experienced significant disease progression, leading to substantial morbidity. This case aligns with other reports of invasive aspergillosis in the brain, where mortality rates remain high despite advances in antifungal therapy.

This case underscores the complexities of diagnosing and managing fungal sinusitis, especially in immunocompromised patients. The overlapping clinical presentation with other conditions, such as temporal arteritis, may delay accurate diagnosis and appropriate treatment. A high degree of suspicion is needed to diagnose fungal sinusitis, and timely initiation of appropriate antifungal therapy and surgical intervention can improve outcomes. Further studies are needed to better understand the pathogenesis and optimal management of fungal sinusitis.

## Figures and Tables

**Figure 1 jof-10-00788-f001:**
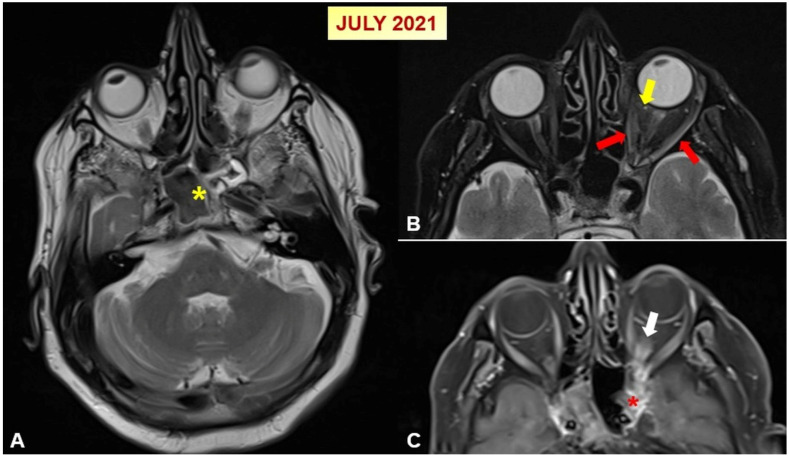
(**A**) An axial T2-weighted orbital MR image shows mucosal thickening in the sphenoid sinus and hypointense areas in the center (yellow asterisk). (**B**,**C**) Orbital MR sections in axial T2-weighted and post-contrast fat-suppressed T1-weighted imaging reveal inflammatory signal alterations in the retroconal adipose tissue (yellow arrow), along with thickening and enhanced contrast uptake in the extraocular muscles (red arrows) and optic nerve (white arrow). Additionally, there is contrast enhancement observed in the left cavernous sinus (red asterisk), suggesting Tolosa–Hunt syndrome or orbital pseudotumor in the left orbit.

**Figure 2 jof-10-00788-f002:**
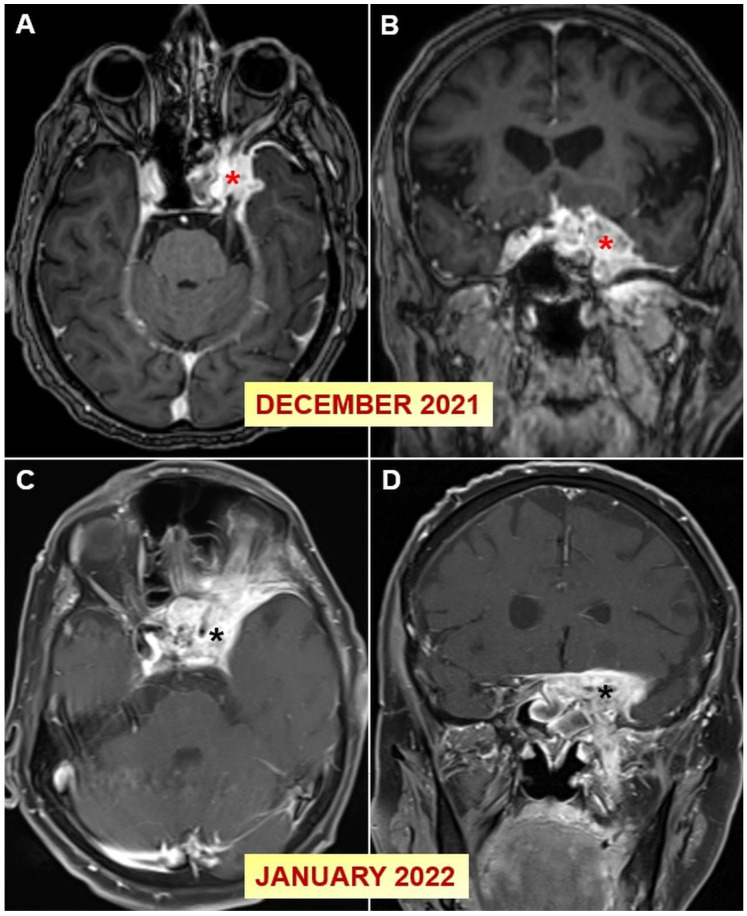
(**A**–**D**) Axial and coronal post-contrast fat-suppressed T1-weighted orbital MR images conducted one month apart demonstrate significant progression of the contrast-enhancing lesion encircling the left optic nerve and left internal carotid artery within the left cavernous sinus (red and black asterisks).

**Figure 3 jof-10-00788-f003:**
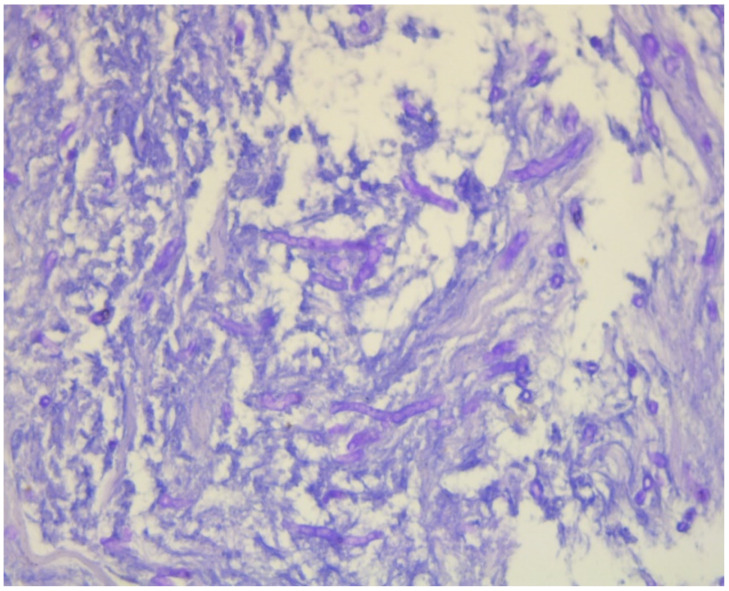
Fungus image with septate hyphae in PAS staining.

**Figure 4 jof-10-00788-f004:**
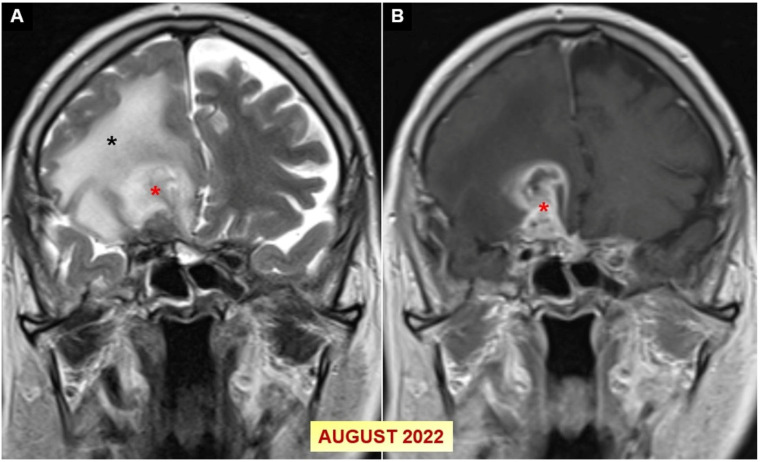
(**A**,**B**) Coronal T2-weighted and post-contrast fat-suppressed T1-weighted cranial MR images conducted in August 2022 reveal peripheral contrast-enhancing foci (red asterisks) encircled by extensive gyral edema (black asterisk) in the right frontal lobe.

## Data Availability

Data are contained within the article.

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
