# Peer review of "Invasive Aspergillosis with Cavernous Sinus Thrombosis Following High-Dose Corticosteroid Therapy: A Challenging Case of Rhino-Orbital-Cerebral Mycosis"

_jof, 2024, doi:10.3390/jof10110788_

Round 1
Reviewer 1 Report
Comments and Suggestions for Authors
This article is an interesting case about fungal rhino-orbital-cerebral mycosis with Cavernous sinus thrombosis(Li DM, Shang PP, Zhu L &. GS de Hoog. Rhino-orbital-cerebral mycosis and cavernous thromboses. Eur J Inflamm. 2014; 12:1-10.),rather than fungal sinusitis. Despite the complexity of the condition, it is largely due to improper diagnosis and treatment, or insufficient understanding of fungal infections(Akarapas C, Wiwatkunupakarn N, Sithirungson S, Chaiyasate S. Anticoagulation for cavernous sinus thrombosis: a systematic 372 review and individual patient data meta-analysis. Eur Arch Otorhinolaryngol. Published online September 23, 2024. ).
Comments on the Quality of English LanguageIt's not as good as what can be published.
Reviewer 2 Report
Comments and Suggestions for Authors
Overall, the reports demostrated an interesting case with detailed description. However, I have several suggestions.
1. The role of COVID-19 in this case should be limited. Please remove the COVID-19 associated content.
2. The structure of this case report should be present in introduction, case report and discussion. Please merge the current method and result section into case report.
3. The abstract needs revision and did not need the structred abstract.
Round 2
Reviewer 1 Report
Comments and Suggestions for Authors
The manuscript is well revised.
Reviewer 2 Report
Comments and Suggestions for Authors
The authors response well.